# A Novel Strategy to Improve Cloud Stability of Orange-Based Juice: Combination of Natural Pectin Methylesterase Inhibitor and High-Pressure Processing

**DOI:** 10.3390/foods12030581

**Published:** 2023-01-29

**Authors:** Wanzhen Zhang, Yantong Li, Yongli Jiang, Xiaosong Hu, Junjie Yi

**Affiliations:** 1Faculty of Food Science and Engineering, Kunming University of Science and Technology, Kunming 650500, China; 2Yunnan Engineering Research Center for Fruit & Vegetable Products, Kunming 650500, China; 3International Green Food Processing Research and Development Center of Kunming City, Kunming 650500, China; 4College of Food Science and Nutritional Engineering, China Agricultural University, Beijing 100083, China

**Keywords:** orange juice, high-pressure processing, cloud stability, pectin methylesterase, pectin methylesterase inhibitor, pectin

## Abstract

This study investigated the prospect of producing cloud-stable orange-based juice by combining high-pressure processing (HPP) with a natural kiwifruit pectin methylesterase inhibitor (PMEI) during chilled storage. Kiwifruit is rich in a PMEI, which greatly improves the cloud loss caused by the pectin methylesterase (PME) demethylation of pectin. The results show that the cloud loss of orange juice occurred after 3 days, while the orange–kiwifruit mixed juice and kiwifruit puree were cloud stable during 28 days’ storage. Although, the kiwifruit puree contained larger particles compared to the orange juice, its higher viscosity and solid-like behavior were dominant, improving the cloud stability of the juice systems. In addition, the particle size distribution and rheological properties were highly related to PME activity, PMEI activity, and pectin characterization. The kiwifruit PMEI showed higher resistance to HPP and storage time than PME. More water-solubilized pectin fractions with a high molecular mass were found in the kiwifruit puree, leading to its high viscosity and large particle size, but a more chelator-solubilized pectin fraction with a low esterification degree was observed in the orange juice, resulting in its cloud loss. In general, the outcome of this work provides a novel strategy to improve the cloud stability of orange-based juices using natural PMEIs and nonthermal processing technologies.

## 1. Introduction

Freshly squeezed orange juice is one of the most popular not-from-concentrate (NFC) juices, and market demand is growing fast due to its health benefits and outstanding mouthfeel [1,2]. Over the past three years, global orange juice production has increased year by year, with both the annual production and consumption remaining above 1.5 million metric tons [2]. However, the shelf life of NFC juices can be greatly affected by some crucial factors, such as cloud loss [3]. Therefore, it is important to extend the shelf life of NFC juices while minimizing impacts on sensory and nutritional qualities.

High-pressure processing (HPP) is an attractive non-thermal preservation technology to fulfill consumer demand due to its excellent pasteurization effect and the maximizing retention of a fresh-like organoleptic quality and nutritional value [4]. Nevertheless, it cannot effectively inactivate the pectin methylase (PME) responsible for pectin degradation and the subsequent cloud loss in orange juices [5]. PME triggers a de-esterification reaction that breaks down pectin to produce pectate, which interacts with calcium ions to induce gelation and reduce cloud stability [6]. Croak et al. [7] reported that the addition of PME to orange juice caused the aggregation of cloud particles within a few minutes. Therefore, the inhibition of PME activity is essential for the stability of the colloidal suspension systems in cloudy orange juices. Some researchers have investigated the effect of inhibitors from natural origins on PME activity, such as epigallocatechin gallate and pectin methylesterase inhibitor (PMEI), which was first identified in ripe kiwifruit [8]. Therefore, kiwifruit has been reported to be an excellent source of natural PME inhibitors, inhibiting PME by forming a reversible non-covalent 1:1 complex with plant PME [9]. The crystal structure complex of the tomato PME–kiwifruit PMEI suggests that the inhibitor covers the active site cleft of PME, thus preventing PME from continuing to react with pectin [10]. However, how clean-label ingredients with natural PMEIs prevent cloud loss and improve pectin characteristics in orange juice is rarely reported.

In addition to enzymatic behavior (e.g., PME-catalyzed hydrolysis), the previous studies found that pectin might undergo non-enzymatic reactions during processing and storage, which could also result in the interconversion of pectin fractions and cloud loss [11]. It has been reported that pectin characterization (e.g., molecular mass, monosaccharide composition, and esterification degree) is one of the main impact factors on rheological properties and particle size, thereby affecting the turbidity stability of the system. Moreover, the osmotic network structure formed between pectin and metal ions can improve the rheological properties, such as viscosity and elastic modulus [7]. The contribution of different pectin fractions to the cloud stability of the juice varied. High relative molecular mass polysaccharides are more likely to precipitate in turbid juice systems and establish intermolecular interactions. Pan et al. [12] found that the highest apparent viscosity of the chelator-solubilized pectin fraction (CSF) in melon juice played an important role in cloud stability [12]. The backbone of pectin consists of D-galacturonic acid condensed with an α-(1–4) glycosidic bond, with neutral sugar molecules, such as galactose (Gal), rhamnose (Rha), and arabinose (Ara) in the side chains. The differences in the content of the neutral sugars in the side chains can cause structural differences in pectin, thus altering the behavior of pectin solubility, thickening, and gelation [13]. As pectin is remarkably correlated with cloud stability, few studies have revealed the effect of HPP combined with the natural PMEI from kiwifruit on the pectin interconversion pattern in juice systems from the perspective of enzymatic reactions. 

The study aims to improve the cloud stability of HPP orange juice using kiwifruit puree and exploring the improvement mechanism by investigating the colloidal characteristics, including PME/PMEI activity, particle size distribution, rheological properties, and pectin characterizations. The results of this work might provide a novel strategy to improve the cloud stability of orange juice using natural and clean-label ingredients and processing technology.

## 2. Materials and Methods

### 2.1. Materials and Reagents

Oranges (*Citrus sinensis* L. *Osbeck*) and gold kiwifruits (*Actinidia chinensis Planch*) without any signs of mechanical damage or fungal decays were purchased from a local market (Kunming City, Yunnan Province, China). Galacturonic acid (GalA) was purchased from BoRui Saccharide (Yangzhou, China). Methanol and acetonitrile were purchased as high performance liquid chromatography (HPLC) grade reagents from Sigma-Aldrich Chemical Co., Ltd. (Shanghai, China) Alcohol oxidase and pectin were purchased from Macklin Biochemical Co., Ltd. (Shanghai, China). All other chemicals and reagents were of analytical grade and purchased from Sinopharm Chemical Reagent Co., Ltd. (Shanghai, China).

### 2.2. Sample Preparation

Oranges were peeled and pressed using a juicer (JYZ-E16, Joyoung Co., Ltd., Zhejiang, China) to squeeze out the cloudy orange juice. Kiwifruits were blended using a mixer (Joyoung JYL-C051, China) to obtain puree after peeling and deseeding. The cloudy orange juice obtained was divided into two parts: one was used as a control sample and called orange juice; the other was thoroughly mixed with kiwifruit puree (25%, *v*/*v*) and referred to as mixed juice. The addition ratio of kiwifruit puree was selected based on the sensory evaluation preliminary study (data not shown). 

The obtained fresh juices were first homogenized under room temperature (20 °C) at 20 MPa for 3 min using a high-pressure homogenizer (GJJ-0.06/70 MPa; Shanghai Noni Light Industrial Machinery, Shanghai, China). After homogenization, all samples were transferred into polyethylene bottles. All samples were processed using HPP, which was performed at 20 °C with a piece of high-pressure equipment (XC-LF3AH, Jiangmen Xiecheng Machinery Co., Ltd., Jiangmen, China). The pressure boosting rate was about 15 MPa/s, and the pressure reached 600 MPa in about 40 s. During processing, the pressure was held for 5 min at 600 MPa. Afterwards, the pressure was released immediately after 10 s. After HPP, all juices were stored in the cooling room at 4 °C for 28 days. Juices were sampled at 0, 3, 7, 14, 21, and 28 days, respectively. Before sampling, a visual appearance was observed to indicate the cloud stability of samples during storage. In addition, the physicochemical properties, rheological properties, particle size distribution (PSD), and activities of pectin methylesterase (PME) and its inhibitor (PMEI) were analyzed immediately. Finally, samples were transferred to plastic centrifuge tubes, frozen in liquid nitrogen, and stored at −40 °C for other quality attributes analysis.

### 2.3. Physiochemical Properties

The physicochemical properties of juices, including pH, total soluble solids (TSS), titratable acid (TA), and color properties, were performed according to our previous procedure [14]. Measurements of pH and TSS were carried out using a pH meter (FE28-Standard, Mettler Toledo, Zurich, Switzerland) and a digital refractometer (TD-45, Jinkelida, Beijing, China) at room temperature, respectively. TA was analyzed using an automatic potentiometric titrator (907 GPD Titrino, Metrohm, Herisau, Switzerland) and calculated using Equation (1). All assays were performed in triplicate.
(1)$$\mathrm{TA}\left(\%\right)=\frac{{C\;\times \;V}_{2}\;\times \;K}{V_{1}}\;\times \;\frac{V_{0}}{W}\;\times \;100$$
where *C* represents sodium hydroxide solution concentration (0.1 mol/L); *W* is the total juice sample weight (g); *V*_2_ means the used NaOH volume (mL); *V*_1_ suggests the juice sample volume used (mL); *V*_0_ signifies the total volume of juice (mL); and *K* is the citric acid conversion factor (0.064).

Color values (*L*^∗^, *a*^∗^, and *b*^∗^) were measured using a colorimeter (Agera, Hunter Associate Laboratory, Inc., Fairfax, VA, USA). The total color difference (Δ*E*^∗^) was calculated as Equation (2): (2)$${\Delta E}^{*}=\sqrt{{[(L}^{*}{\;-\;L}_{0}^{*}{)}^{2}{+(a}^{*}{-a}_{0}^{*}{)}^{2}{+(b}^{*}{-b}_{0}^{*}{)}^{2}]}$$
where *L*^∗^ represents brightness, *a*^∗^ is red-green values, *b*^∗^ signifies yellow-blue value, and the variables with subscript ‘0′ means the initial values of non-stored juices. All measurements were repeated in triplicate.

### 2.4. Rheological Properties

The rheological behaviors of each sample were determined at 25 °C using a modular compact rheometer (MCR 102, Anton Paar, Graz, Austria) fitted with a Couette-geometry sensor (concentric cylinder, Graz, Anton Paar CC27) and a four-bladed vane geometry (ST22-4V-40) according to a reported study with slight modifications [15]. Each juice sample was adequately mixed and preheated to 25 °C. Then, 30 mL of the sample was slowly poured into the cylinder, and then the spindle was immersed under the sample. The viscosities were recorded as a function of shear rate from 0.1 to 500 s^−1^ to measure the apparent viscosity. The frequency sweep of juice sample was carried out from 3 to 300 rad/s at 25 °C under an oscillation strain of 1%. The storage modulus (*G′*, Pa) and loss modulus (*G″*, Pa) were recorded as a function of frequency. The frequency dependence of *G′* and *G″* was assessed using the power law models in the method described in the previous study [16]. All experiments were conducted in triplicate.
(3)$$G^{\prime }\;=K^{\prime }\;•\;\omega \;n^{\prime }$$
(4)$$G^{\prime \prime }\;=K^{\prime \prime }•\;\omega \;n^{\prime \prime }$$
(5)$$\eta =k\;•\;\gamma \left(n-1\right)$$
where *K′* and *K″* are constants; *ω* is the angular frequency; *n′* and *n″* refer to the frequency exponent; *η* is the apparent viscosity; *k* is the consistency coefficient; *γ* is the shear rate; and *n* is the flow behavior index.

### 2.5. Particle Size Distribution

The particle size distribution (PSD) of each sample was evaluated using laser grain size diffraction (Mastersizer 3000, Malvern Instruments Limited, Worcestershire, UK) as described by the reported procedure with slight modifications [16]. After mixing thoroughly, the juice samples were slowly added into a stirring beaker filled with 500 mL distilled water for dispersion until the polarization intensity differential scattering reached 8%. The parameters (D [4, 3] and D [3, 2]) were obtained after three times of repeating.

### 2.6. Enzyme Activities

#### 2.6.1. PME Activity

The PME extraction and activity assays were performed according to Yi et al. [14] with slight modifications. Juice (50 mL) was centrifugated at 10,000× *g* for 1 h at 4 °C (LTNX-6000 centrifuge, Thermo Fisher Scientific, Waltham, MA, USA). The precipitate was collected and then mixed end-over-end in the extraction buffer (0.2 mol/L Tris-HCl with 1.0 mol/L NaCl, pH 8.0) with a ratio of 1:1.3 (*w*/*v*) for 4 h at 4 °C. After that, the mixture was centrifuged twice at 10,000× *g* for 30 min at 4 °C. The resulting supernatant was used as PME crude extract. The extraction was carried out twice. Crude extracts of PME were preserved in the buffer containing 0.2 mmol/L Tris-HCl and 1 mol/L NaCl (pH 8.0). The activity of PME was measured using an automatic potentiometric titrator (Metrohm, Herisau, Switzerland) at pH 7 and 25 °C. The crude PME extract was mixed with 0.35% (*w*/*v*) apple pectin solution (0.117 mol/L NaCl and 0.01 mol/L NaOH). The PME activity was represented by one unit (U) as the amount of enzyme producing 1 µmol of carboxyl groups per minute. All assays were conducted in triplicate.

#### 2.6.2. PMEI Inhibitory Activity

The PMEI extraction and inhibition assays were performed according to a reported study with slight modifications [17]. Samples were collected and homogenized with 7.5% polyvinyl polypyrrolidone (PVPP) (1:1, *w*/*v*) at 4 °C. After centrifugation for 20 min (20,000× *g*, 4 °C), the supernatant was isolated and collected. The insoluble particles in the supernatant were removed using filtration, and the pH was adjusted to 6.5. The resulting liquid was PMEI crude extract. The PMEI activity was expressed as the inhibition rate of PME activity. All assays were conducted in triplicate.

### 2.7. Pectin Characterization

#### 2.7.1. Alcohol-Insoluble Residue Extraction and Fractionation

The alcohol-insoluble residue (AIR) was extracted and successively fractionated into water-solubilized pectin fraction (WSF), chelator-solubilized pectin fraction (CSF), and sodium carbonate-solubilized pectin fraction (NSF), as reported in our previous study [15]. The neutral sugar composition and molar mass distribution of AIR, WSF, CSF, and NSF were measured, as previously described [3].

The procedures of pectin fraction extraction were followed by the approach described in our previous research [15]. Juice samples (100 g) were homogenized in 500 mL of 95% ethanol for 30 min at 4 °C. The ethanol and juice mixture were filtered to separate them, and then it was homogenized again in 95% ethanol. The mixture was then screened once more. Acetone was used to homogenize the AIR. The AIR fraction was produced after final vacuum filtration and vigorous mixing. Based on solubilization in various solutions, the AIR was successively segregated into three parts: NSF, CSF, and WSF. All fractions were lyophilized and kept until analysis in a desiccator over phosphorus pentoxide.

#### 2.7.2. Degree of Esterification Analysis

The degree of esterification (DE) value was measured using Fourier transform infrared spectra using previous method [12]. The peaks at 1749 cm^−1^ (the number of carboxyl groups) and 1617 cm^−1^ (the number of free carboxyl groups) were recorded. The equation used to determine DE values was as follows:(6)$$DE=\frac{A_{1749}}{A_{1749}+A_{1617}}\times 100\%$$

#### 2.7.3. Molar Mass Distribution Measurement

The molecular weight of the polysaccharides was determined using high performance liquid chromatography [18]. The samples and standards were accurately weighed and prepared in a 5 mg/mL solution and then centrifuged at 12,000× *g* for 10 min. The supernatant was filtered through a 0.22 μm microporous membrane and then transferred to a 1.8 mL injection vial. The molecular weight of each sample was calculated from the standards. Chromatographic column: BRT105-104-102 tandem gel column (8 mm × 300 mm); mobile phase: 0.05 mol/L sodium chloride solution; flow rate: 0.6 mL/min; column temperature: 40 °C; injection volume: 20 μL; detector: RI-10A for differential detection. 

#### 2.7.4. GalA Content Measurement

The measurement of GalA content was carried out according to our previous experiments [15]. The AIRs, WSFs, CSFs, and NSFs were hydrolyzed using sulfuric acid containing 0.0125 mol/L sodium tetraborate and heated at 100 °C for 10 min. After cooling down, the 3-hydroxybiphenyl (0.15%, *w*/*v*) was dissolved in NaOH (0.5%, *w*/*v*). Then, the absorbance of mixture was measured at 520 nm using an ultraviolet-visible spectrophotometer (T9CS, Persee, China). A standard curve was created using GalA at concentrations between 10 and 120 μg/mL. Each sample was analyzed in triplicate.

### 2.8. Data Analysis

All of the sample treatments and experimental determinations were repeated in triplicate. All experimental data were analyzed with mean value ± standard deviation. The statistical significance of the difference was evaluated using Tukey’s test at a 95% significance level on Statistical Product and Service Solutions (SPSS, 20.0 statistics software, IBM, Armonk, NY, USA).

## 3. Results and Discussion

### 3.1. Visual Appearance

The visual appearance of juices on the shelf directly affects the consumers’ purchasing behavior, thus it is essential to evaluate the appearance changes in the storage period of juices [2]. As shown in Figure 1, visually, HPP did not cause clear changes in the turbidity homeostasis of juices, either for orange juice, mixed juice, or kiwifruit puree. The cloud in the orange juice sample was noticeably thinner after 3 days of storage. As for the 28th-day orange juice sample, it is seen that the suspended substances (e.g., pectin and protein) separated from the liquid phase, leaving a yellowish, unacceptable serum and the resulting “water-like clarity”, which was generally called cloud loss [7]. After adding kiwifruit puree, the mixed juice showed better cloud stability compared to the orange juice. Cloudy fruit and vegetable beverages are generally considered to be a complex serum of pectin, sugar, organic acids, and salts which contains a colloidal dispersion of charged particles [19]. Although juices appear well cloudy when freshly squeezed, many chemical and biological changes occur during storage, including molecular interactions, molecular polymerization, and pectin cross-linking. The above changes critically affect the consumer acceptance of juices [20]. In this research, the result agreed with the result of Yi et al. [21,22] and Li et al. [23]; after the addition of kiwifruit puree, no clear sedimentation appeared in the mixed juices during storage.

Color change is an important visual indicator of juices during storage. It was generally considered that the color difference could be visible for consumers when the Δ*E** value exceeds 3 [23]. It can be seen from Table 1 that the addition of golden kiwifruit puree did not affect the characteristic yellow color (*b**) and *L** of the orange juice. Mixed juice represented the lowest Δ*E** value, indicating that its color was more stable than orange juice and kiwifruit puree. The color parameters of the kiwifruit puree manifested that the kiwifruit puree got slightly dark, which may be the cause of ascorbic acid degradation [4]. Yi et al. [24] pointed out that HPP had little effect on the presentation of the pigment molecules in fruit and vegetable juices, while most of the browning changes during storage were caused by endogenous enzymatic activity and non-enzymatic reactions [25].

### 3.2. Physicochemical Properties

#### 3.2.1. pH, TSS, and TA Values

As shown in Table 1, the impact of HPP on the pH, TSS, and TA values of the juices is negligible. The low pH value of kiwifruit (3.48) resulted in a decrease in the pH from 4.35 (orange juice) to 3.89 (mixed juice), which was associated with its rich ascorbic acid content [26]. As a consequence, among the three sample groups, kiwifruit puree had the highest TA value, followed by mixed juice. Likewise, the abundance of sugar content in kiwifruit (11.60 ^◦^Brix) led to an increase in TSS values from 8.70 °Brix (orange juice) to 9.60 °Brix (mixed juice). In addition, the pH, TA, and TSS values of the juices were stable throughout the storage period for all groups. The orange and kiwifruit materials used in this study had high maturity, which may be the reason for the stable content of the soluble solids in the juices during storage.

#### 3.2.2. Particle Size Distribution

The PSD and particle diameters of orange juice, mixed juice, and kiwifruit puree are shown in Figure 2 and Figure 3. A monomodal particle size distribution for all juice samples can be observed (Figure 2). It might be attributed that all juice in the study were homogenized, which contributed a more uniform particle distribution to the juice systems. Juice after homogenization was reported to show a monomodal particle size distribution in cloudy apple juice [22] and orange juice with guar gum added [27]. When comparing different juice systems, the narrowest particle size distribution was observed in the orange juice, while a wider particle size distribution was exhibited in the kiwifruit puree. The particle size distribution of mixed juice was similar to that of orange juice. It seems that the effect of orange juice on the particle size of mixed juice was more dominant than that of the kiwifruit puree. Trends could be found on the D [4, 3] and D [3, 2] values (Figure 3A,B). The lowest value of D [4, 3] and D [3, 2] (126.94 and 26.96 μm, respectively) was observed in orange juice, but the highest value of D [4, 3] and D [3, 2] (220.62 and 57.22 μm, respectively) was showed in the kiwifruit puree. As mixed juice was homogenized with orange juice and kiwifruit puree, their D [4, 3] and D [3, 2] values fell in the range between orange juice and kiwifruit puree (160.92 and 29.07 μm, respectively). However, the particle size of mixed juice was closer to orange juice compared to the kiwifruit puree. 

HPP and storage had no significant effect on the particle size of the mixed juice and kiwifruit puree (*p* > 0.05). However, the particle size of the orange juice was more affected by HPP and prolonged chilled storage. A decreased particle size of the orange juice was observed after HPP, but an increase trend in the particle size occurred in orange juice during storage. As the orange juice was cloudy juice with some fine pulp and fiber, possible granule entanglement and/or separation might lead it to being sensitive to processing and storage [28]. It seems that the addition of a high viscous puree could largely enhance the resistance of cloudy orange juice to the impact of HPP and storage. In a cloudy apple-based juice system, a similar trend was also observed: apple juice mixed with kiwifruit puree could increase the stability of the particle size during high-pressure processing and refrigerated storage [21].

#### 3.2.3. Rheological Properties

Figure 4A–C shows the change curve of apparent viscosity in all juices with the shear rate, and their power law parameters are shown in Table 2. The apparent viscosity of orange juice, mixed juice, and kiwifruit puree decreased with an increasing shear rate, exhibiting the shear thinning phenomenon and pseudoplastic fluid characteristics. It was also supported by the data presented in Table 2.

The turbidity stability of fruit juices is closely related to viscosity [29]. As shown in Figure 4D, the kiwifruit puree exhibited the highest viscosity (2019.35 mPa·s), followed by mixed juice (232.42 mPa·s), and orange juice showed the lowest viscosity (14.24 mPa·s). It indicated that adding kiwifruit could significantly raise the viscosity of mixed juice (*p* < 0.05). In addition, HPP had less impact on the viscosity of the juice samples, particularly for the mixed juice and kiwifruit puree (*p* > 0.05). During storage, the viscosities of all samples remained stable. Karahman et al. [30] found that the viscosity of apple–carrot juice remained stable during 21-day storage. As known, the viscosity of fruit juice is considered to be closely related to particle size and composition [31]. During storage, the particles of the mixed juice and kiwifruit puree increased the cohesive energy of the dispersed phase, thereby controlling their viscoelastic behavior [32]. The viscosity remained stable as a function of storage time due to the presence of stable particles [33]. 

The frequency sweep showed the change in the storage modulus *G′* and loss modulus *G″* of fruit juice at different angular frequencies. In addition, the loss angle tangent (tan *б* = *G″*/*G′*) indicated the relative fluidity of the matrix [31]. In other words, tan *б* can reflect the viscoelasticity of juice [31]. As shown in Figure 5A–C, different viscoelastic behavior was observed for the three groups. However, HPP and storage had less effect on the viscoelastic behavior of all juice and puree systems (*p <* 0.05). In orange juice, the loss modulus was higher than the storage modulus (*G″* > *G′*; tan *б* > 1), indicating poorly ordered particles, viscous deformation, and system liquid characteristics [28]. However, at low-frequency shearing, tan *б* < 1 was observed in the mixed juice and kiwifruit puree. When increasing the shear frequency, a higher loss modulus but a lower storage modulus (tan *б* > 1) could be found in the mixed juice and kiwifruit puree. It demonstrated there was a shift from solid-like behavior to viscous-like behavior of the mixed juice and kiwifruit puree. Although similar behavior was observed for both the mixed juice and kiwifruit puree, the kiwifruit puree had a much higher storage modulus than the mixed juice. Wang et al. [34] found a similar phenomenon in kiwifruit juice. It might be because the presence of larger particles in the kiwifruit puree matrix could control the viscoelastic behavior of the suspended particles, thus resulting in a higher modulus [28,31]. Meanwhile, the angular frequency of the critical gel point in the kiwifruit puree increased at tan *б* = 1, which might be because of the high content of solute particles in the system [31]. The kiwifruit puree became thicker with decreased fluidity and increased viscoelasticity. Consequently, the particles in the kiwifruit puree did not easily settle, thereby improving its suspension stability. This phenomenon was also reflected by visual changes: the orange juice showed the strongest fluidity and fast sediment, but the mixed juice and kiwifruit puree flowed much slower and exhibited higher cloud stability.

### 3.3. Enzyme Activity

Figure 6 demonstrates the changes in the activity of PME and its inhibitor in orange juice, mixed juice, and kiwifruit puree. The PME activity of the crude extracts from orange juice, mixed juice, and kiwifruit puree was 1.92, 0.57, and 0.04 U/mL of juice, respectively. Koh et al. [20] found that the PME activity of orange juice was a 1.2 U/mL of juice sample, which was consistent with our work. As shown in Figure 6B, PMEI activity was not detected in orange juice. However, the PMEI inhibition ratio was found at 51 and 33% in the kiwifruit puree and mixed juice, respectively. As shown, PME is one of the main endogenous enzymes in oranges, but it was found less in kiwifruit. However, according to Jolie et al. [9], PMEI, as the natural enzyme inhibitor in kiwifruit, was absent in oranges. The low PME activity in kiwifruit might be because of the existence of the natural PMEI in kiwifruit [17]. Thus, the addition of kiwifruit decreased the residual PME activity of mixed juice to 30.02% (Figure 6A).

Figure 6A shows that HPP reduced the PME activity by 60% in orange juice. Tian et al. [8] observed that the residual PME activity in orange juice decreased to 42.41% at a pressure of 600 MPa, which was consistent with our research. The partial inactivation of PME by HPP might be because of the possible folding and unfolding of the PME structure during the pressure increase and release period [6,17]. Nevertheless, HPP had no impact on the PME and PMEI of the mixed juice and kiwifruit puree (*p >* 0.05). It seems that although HPP has an inactivation effect on the endogenous PME activity of oranges, its impact on PMEI was limited [9]. This was attributed to the pressure-stable intermediate state of PMEI induced by HPP [17]. In addition, the synergy of natural PMEI in kiwifruit showed a more dominant influence on the PME residual activity in the mixed juice. In other words, the PMEI from kiwifruit covered the putative active site cleft of PME, thereby possibly forming a PMEI/PMEI combination [35]. The combination might have an enhanced resistance to high pressure in the mixed juice compared to PME. 

During storage, the residual PME activity of orange juice showed an upward trend, whereas mixed juice exhibited a downward tendency (Figure 6A). However, the PMEI inhibition capacity in the kiwifruit puree and mixed juice remained stable throughout the entire storage period (Figure 6B). Welti-Chanes et al. [36] noted that PME activity in Valencia orange juice increased during storage due to the presence of stable PME isomers. Previous studies have pointed out that this phenomenon might be related to enzyme properties, applied pressure levels, and environmental aspects [6]. For example, the interaction between PME and cations as well as its isozymes should be taken into account [37]. The decrease in PME residual activity in mixed juice might be related to the changes in the PMEI/PMEI complex [17]. During storage, the PMEI/PMEI complexes in mixed juice might be dissociated [17,35]. The released PMEI could continue to bind to the residual PME, leading to less PME activity that could be detected. In general, kiwifruit puree addition might be an innovative strategy for reducing the endogenous PME activity in fruit-based juices, although future work needs to be carried out to further demonstrate the mechanism hypothesis on the effect of HPP and storage time on purified PME, the PMEI, and their complex.

### 3.4. Pectin Characterization

Pectin characteristics play an important role in cloud loss by forming calcium pectate complexes and precipitating insoluble particles under PME catalysis. As for pectin characteristics, the GalA content is normally used to describe the pectin content as it is derived from homogalacturonan and the RG-I main chain. To better understand the possible sedimentation mechanism of orange-based juices, the pectin characterization of different pectin fractions in three juice/puree systems and a pectin-related schematic diagram of suspension stability are illustrated in Figure 7 and Figure 8, respectively.

As shown in Figure 7A, the highest GalA content of AIR was observed in the kiwifruit puree (5.87 mg/mL of juice), followed by mixed juice (2.04 mg/mL of juice), and the lowest GalA content was detected in orange juice (2.40 mg/mL of juice). The high content of GalA has been reported in kiwifruit [38]. It demonstrates that the pectin content of kiwifruit puree was significantly (*p* < 0.05) higher than orange juice, which was beneficial to enhancing the viscosity of the mixed juice and kiwifruit puree. In addition, HPP had no significant impact on the GalA contents of all samples (*p >* 0.05). During storage, the GalA content of AIR in orange juice and mixed juice showed a decreasing trend but remained stable in the kiwifruit puree. The decrease in the GalA content of AIR in orange-based juice during storage might be related to the orange pectin backbone depolymerization [11]. As for different pectin fractions, fresh orange juice showed the highest GalA content of CSF and NSF compared to other juices, while the fresh mixed juice and kiwifruit puree exhibited a higher GalA content of WSF than orange juice. It seems that orange juice contained higher CSF and NSF, but the kiwifruit puree and mixed juice contained higher WSF. As illustrated in Figure 8, CSF and NSF influenced stability properties by forming networks with calcium bridges and trapping molecules, such as protein and phenol, in the orange juice [12,15]. This might relate to the faster semination of orange juice than the other two juice systems (Figure 1). On the other hand, the mixed juice and kiwifruit puree contain homogeneously dispersed WSF molecules. As the proportion of hydrated molecules increased, the velocity pattern of the liquid distorted, and the viscosity increased [13]. As WSF could contribute high water solubility and high viscosity to the juice [11,15], it might explain why the kiwifruit puree and mixed juice exhibited higher viscosity than orange juice (Figure 4D).

Figure 7B shows that the molecular mass of AIR in the kiwifruit puree (142.52 kDa) was significantly higher (*p <* 0.05) than that of orange juice (10.15 kDa). The higher value of the molecular mass indicated a larger particle size [39]. The results on particle size distribution (Figure 2 and Figure 3) agreed with the data on the molecular mass distribution (Figure 7B): the kiwifruit puree presented the largest particle sizes, while the smallest particle size was found in orange juice (Figure 8). In addition, there was no significant change in the molar mass of pectin fractions during HPP and storage (*p >* 0.05). As known, large particle size would promote cloud loss and phase separation of juice, although mixed juice with larger particles but higher viscosity still showed a higher cloud stability than orange juice. It indicated that the effect of viscosity on cloud stability was more dominant than the particle size. In addition, the solid-like behavior caused by larger particles in the kiwifruit puree matrix could also better control the viscoelastic behavior of the suspended particles, thus resulting in a higher cloud suspension stability. When the juice showed solid-like behavior, the distance between pectin molecules might decrease, thereby promoting intermolecular interactions such as hydrogen bonding [13]. 

The DE value affected the cloud stability through low DE pectin molecules combining with calcium ions or protein to form insoluble large particles (also known as the “egg-box model”) and protein–pectin flocculation, which leads to the aggregation and precipitation of particles [40,41]. Pectin could normally be divided into high methoxyl pectin with DE > 50% and low methoxyl pectin with DE < 50% [42]. Figure 7C shows that the AIR of orange juice and kiwifruit puree were both low DE pectin, with values of 29.35 and 23.75%, respectively. HPP did not significantly affect the DE of all samples (*p >* 0.05). During storage, the DE value decreased in orange juice but remained stable in the mixed juice and kiwifruit puree. As for different pectin fractions, only the DE of WSF was detected, but the DE of CSF and NSF were all below the detection limit in all samples. Similar to that of AIR, the DE of WSF in the kiwifruit puree remained stable during storage, but a significant decrease in WSF’s DE was found in orange juice. The decrease in the DE of orange juice might result from residual PME activity [37]. However, the stable DE of the kiwifruit puree and mixed juice might be attributed to the existence of a natural PMEI. As illustrated in Figure 8, the residual active part of PME was mainly bound to the PMEI, thus weakening the demethylation effect of PME on pectin. Consequently, more stable particle distribution was observed in the mixed juice and kiwifruit puree. In general, changes in pectin characterization and its related enzyme activity were important factors affecting the cloud stability of orange-based juice systems during processing and storage.

## 4. Conclusions

This study investigated the possibility of producing clean-label and cloud-stable orange juice by combining HPP with kiwifruit puree. The turbidity stability of mixed juice was obviously enhanced by kiwifruit puree combined with HPP. During the 28 days of storage, the physicochemical properties, color change, viscosity, and particle size distribution of mixed juices remained stable. The relative residual activity of PME was decreased, and the inhibitory capacity of the PMEI was stable during storage. It was found that changes in pectin characterization were an important factor affecting the cloud stability of orange-based juice systems during processing and storage. Mixed juice contained higher pectin GalA contents and molecular mass than orange juice, particularly for WSF, but more CSF and NSF were detected in orange juice. In addition, the pectin in all juices were low DE pectin, with the DE value of mixed juice staying stable during storage. It might be related to the action of the PMEI in kiwifruit, which could bind PME to weaken the demethylation effect of PME on pectin.

In general, the outcome of this work provides a novel strategy to improve the cloud stability of orange-based juice using natural PMEIs and nonthermal processing technologies. In addition, changes in pectin induced by PME, PMEIs, processing, and storage time were related to the high cloud stability of the fruit/vegetable-based juices. Therefore, future work still needs to be carried out to further demonstrate the mechanism hypothesis on the effect of HPP and storage time on purified pectin, PME, PMEIs, and their complex. 

## Figures and Tables

**Figure 1 foods-12-00581-f001:**
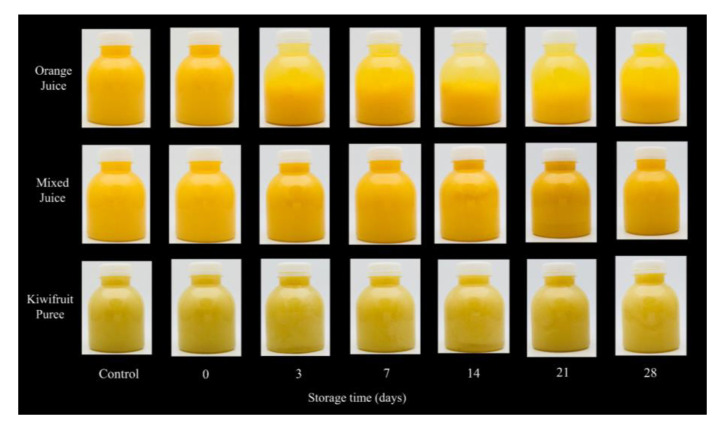
Visual appearance of orange juice, mixed juice, and kiwifruit puree during 28 days of storage at 4 °C.

**Figure 2 foods-12-00581-f002:**
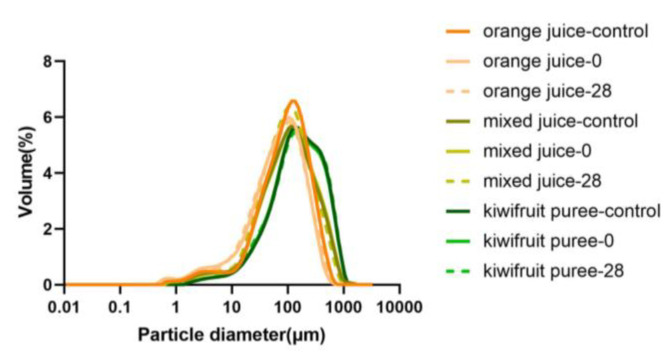
Particle size distribution of orange juice, mixed juice, and kiwifruit puree during high-pressure processing and storage at 4 °C.

**Figure 3 foods-12-00581-f003:**
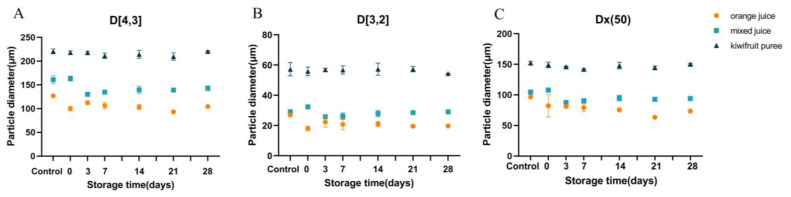
Particle diameters of orange juice, mixed juice, and kiwifruit puree during high-pressure processing and storage at 4 °C. (**A**): volume-based mean diameter (D [4, 3]). (**B**): surface-area-based mean diameter (D [3, 2]). (**C**): mode diameter (Dx (50)).

**Figure 4 foods-12-00581-f004:**
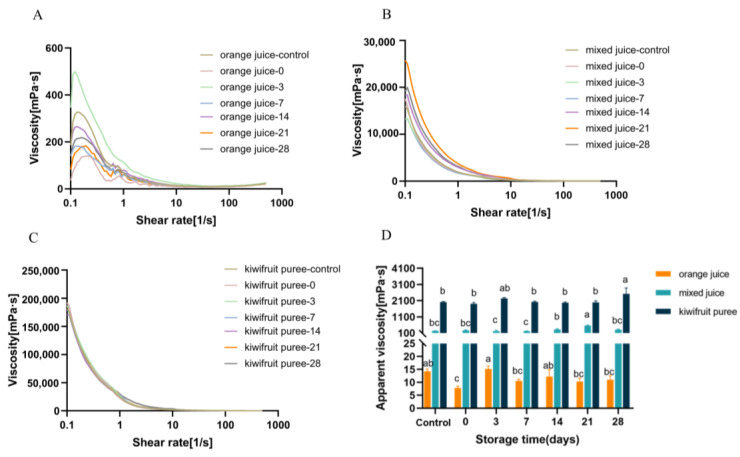
The flow behavior and apparent viscosity of orange juice, mixed juice, and kiwifruit puree during high-pressure processing and storage at 4 °C. (**A**): apparent viscosity of orange juice; (**B**): apparent viscosity of mixed juice; (**C**) apparent viscosity of kiwifruit puree; (**D**): apparent viscosity at the shear rate of 10 s^−1^ of orange juice, mixed juice, and kiwifruit puree. Different letters in the same color of the bar indicate statistically significant differences determined by Tukey’s HSD test (*p* < 0.05).

**Figure 5 foods-12-00581-f005:**
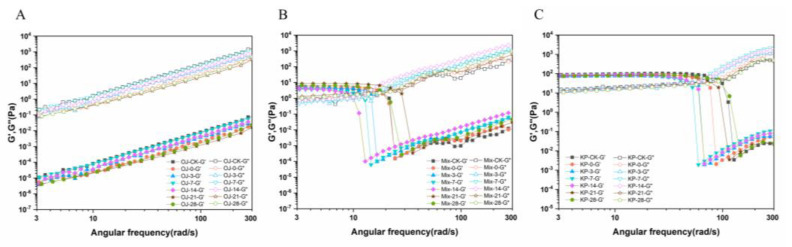
The storage modulus (*G′*) and loss modulus (*G″*) of orange juice (OJ), mixed juice (MJ), and kiwifruit puree (KP) during high-pressure processing and storage at 4 °C. (**A**): storage modulus (*G′*) and loss modulus (*G″*) of orange juice; (**B**): storage modulus (*G′*) and loss modulus (*G″*) of mixed juice; (**C**): storage modulus (*G′*) and loss modulus (*G″*) of kiwifruit puree.

**Figure 6 foods-12-00581-f006:**
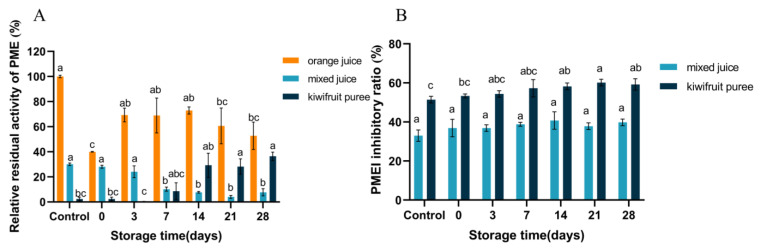
The relative residual activity of PME (**A**) and inhibitory ratio of PMEI (**B**) in orange juice, mixed juice, and kiwifruit puree during processing and storage. PME: pectin methylesterase; PMEI: pectin methylesterase inhibitor. Different letters in the same color of the bar indicate statistically significant differences determined by Tukey’s HSD test (*p* < 0.05).

**Figure 7 foods-12-00581-f007:**
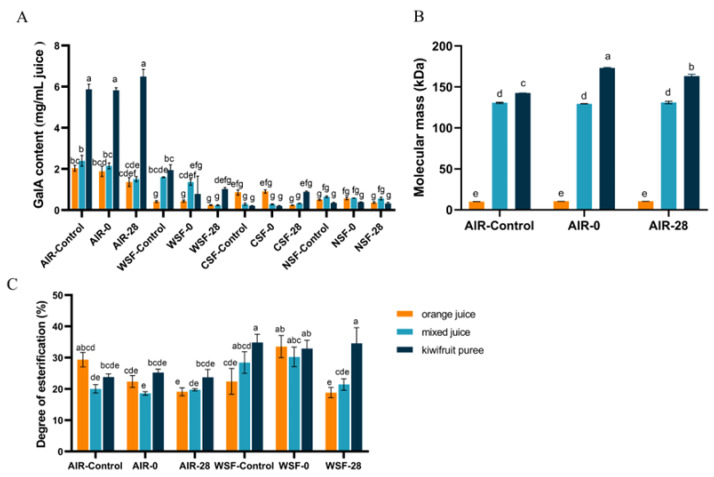
Characterization of different pectin fractions in orange juice, mixed juice, and kiwifruit puree during high-pressure processing and storage. (**A**): GalA content of AIR, WSF, CSF, and NSF in orange juice, mixed juice, and kiwifruit puree. (**B**): molecule mass of AIR in orange juice, mixed juice, and kiwifruit puree. (**C**): degree of esterification of AIR and WSF in orange juice, mixed juice, and kiwifruit puree. GalA: galacturonic acid; AIR: alcohol-insoluble residue; WSF: water-solubilized pectin fraction; CSF: chelator-solubilized pectin fraction; NSF: sodium carbonate-solubilized pectin fraction. Different letters in the same row indicate statistically significant differences determined by Tukey’s HSD test (*p* < 0.05).

**Figure 8 foods-12-00581-f008:**
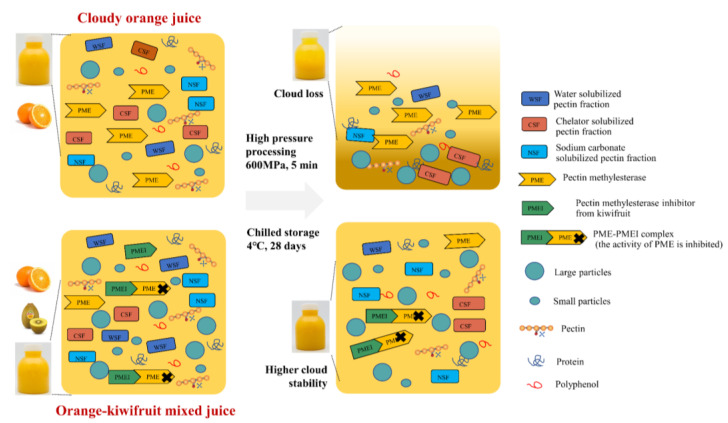
Schematic diagram of suspension stability of orange juice and mixed juice during storage.

**Table 1 foods-12-00581-t001:** Physicochemical properties of orange juice, mixed juice, and kiwifruit puree after high-pressure processing (HPP) during storage for 28 days at 4 °C.

Storage Time (Days)		Color				pH	TSS (°Brix)	TA (%)
		*L^*^*	*a^*^*	*b^*^*	Δ*E*^*^			
Orange juice	Control	58.28 ± 0.23 ^cde^	6.44 ± 0.30 ^a^	51.07 ± 0.52 ^a^	– *	4.35 ± 0.09 ^a^	8.70 ± 0.00 ^a^	0.31 ± 0.00 ^a^
	0	57.97 ± 0.33 ^de^	3.93 ± 0.16 ^cd^	45.68 ± 0.25 ^b^	5.50 ± 0.27 ^a^	4.38 ± 0.03 ^a^	8.50 ± 0. 26 ^ab^	0.28 ± 0.00 ^c^
	3	59.15 ± 0.62 ^bc^	4.81 ± 1.25 ^bc^	45.81 ± 1.29 ^b^	1.23 ± 0.14 ^c^	4.27 ± 0.04 ^b^	8.63 ± 0.12 ^ab^	0.29 ± 0.01 ^bc^
	7	59.72 ± 0.40 ^ab^	4.73 ± 0.08 ^bc^	45.62 ± 0.96 ^b^	1.63 ± 0.22 ^bc^	4.32 ± 0.02 ^ab^	8.47 ± 0.05 ^b^	0.30 ± 0.01 ^ab^
	14	60.59 ± 0.96 ^a^	5.22 ± 0.37 ^b^	46.31 ± 1.19 ^b^	2.98 ± 0.82 ^a^	4.36 ± 0.01 ^a^	8.43 ± 0.05 ^b^	0.31 ± 0.00 ^a^
	21	57.80 ± 0.72 ^e^	3.67 ± 0.16 ^d^	43.64 ± 0.55 ^c^	2.08 ± 0.49 ^b^	4.35 ± 0.01 ^a^	8.43 ± 0.15 ^b^	0.30 ± 0.01 ^a^
	28	59.03 ± 0.67 ^bcd^	4.79 ± 0.04 ^bc^	46.23 ± 0.07 ^b^	1.36 ± 0.07 ^bc^	4.35 ± 0.01 ^a^	8.56 ± 0.05 ^ab^	0.30 ± 0.11 ^a^
Mixed juice	Control	57.94 ± 0.11 ^a^	5.11 ± 0.08 ^cd^	50.07 ± 0.12 ^a^	– *	3.89 ± 0.05 ^a^	9.60 ± 0.00 ^b^	0.45 ± 0.00 ^bc^
	0	56.93 ± 0.57 ^ab^	4.83 ± 0.34 ^d^	48.77 ± 0.66 ^ab^	1.67 ± 0.79 ^ab^	3.93 ± 0.08 ^a^	9.70 ± 0.00 ^a^	0.44 ± 0.01 ^c^
	3	56.19 ± 0.29 ^b^	4.92 ± 0.16 ^d^	47.53 ± 0.47 ^bc^	1.47 ± 0.51 ^ab^	3.90 ± 0.03 ^a^	9.63 ± 0.06 ^ab^	0.48 ± 0.01 ^a^
	7	57.04 ± 1.14 ^ab^	5.30 ± 0.32 ^cd^	46.98 ± 0.38 ^c^	2.13 ± 0.48 ^a^	3.89 ± 0.02 ^a^	9.68 ± 0.05 ^ab^	0.46 ± 0.01 ^bc^
	14	57.42 ± 0.75 ^a^	5.50 ± 0.49 ^bc^	47.66 ± 1.81 ^bc^	2.21 ± 0.63 ^a^	3.92 ± 0.02 ^a^	9.63 ± 0.06 ^ab^	0.45 ± 0.00 ^bc^
	21	57.74 ± 0.24 ^a^	6.08 ± 0.09 ^a^	49.23 ± 0.28 ^a^	1.49 ± 0.16 ^ab^	3.89 ± 0.03 ^a^	9.60 ± 0.00 ^b^	0.47 ± 0.01 ^ab^
	28	57.44 ± 0.26 ^a^	5.92 ± 0.15 ^ab^	49.17 ± 0.29 ^a^	1.29 ± 0.33 ^b^	3.95 ± 0.01 ^a^	9.48 ± 0.22 ^ab^	0.47 ± 0.00 ^ab^
Kiwifruit puree	Control	61.46 ± 0.04 ^b^	−1.46 ± 0.04 ^c^	27.29 ± 0.09 ^b^	– *	3.48 ± 0.02 ^b^	11.60 ± 0.00 ^ab^	0.84 ± 0.04 ^cd^
	0	62.12 ± 0.06 ^a^	−1.16 ± 0.02 ^a^	28.16 ± 0.07 ^a^	1.14 ± 0.11 ^e^	3.54 ± 0.02 ^a^	11.77 ± 0.06 ^a^	0.86 ± 0.12 ^bcd^
	3	59.40 ± 0.27 ^d^	−1.49 ± 0.04 ^c^	26.11 ± 0.54 ^cd^	3.46 ± 0.11 ^bc^	3.44 ± 0.01 ^c^	11.63 ± 0.12 ^ab^	0.78 ± 0.06 ^d^
	7	59.77 ± 0.09 ^c^	−1.46 ± 0.10 ^c^	25.69 ± 0.28 ^d^	3.43 ± 0.17 ^c^	3.44 ± 0.01 ^c^	11.63 ± 0.09 ^ab^	0.93 ± 0.06 ^a^
	14	59.63 ± 0.10 ^cd^	−1.30 ± 0.04 ^b^	26.49 ± 0.31 ^c^	3.01 ± 0.10 ^d^	3.48 ± 0.01 ^b^	11.48 ± 0.13 ^b^	1.03 ± 0.13 ^abc^
	21	58.44 ± 0.14 ^e^	−1.23 ± 0.12 ^ab^	27.59 ± 0.24 ^b^	3.73 ± 0.17 ^a^	3.48 ± 0.01 ^b^	11.56 ± 0.05 ^b^	1.09 ± 0.03 ^a^
	28	58.53 ± 0.06 ^e^	−1.19 ± 0.05 ^ab^	27.62 ± 0.11 ^ab^	3.63 ± 0.07 ^ab^	3.50 ± 0.01 ^b^	11.78 ± 0.15 ^a^	1.06 ± 0.01 ^ab^

Mean ± standard deviation. Different letters in the same row indicate statistically significant differences determined by Tukey’s HSD test (*p* < 0.05). * The non-HPP-treated orange juice, mixed juice, and kiwifruit puree are used as a reference to calculate Δ*E** compared with 0-day samples after HPP.

**Table 2 foods-12-00581-t002:** Power law model parameters of orange juice, mixed juice, and kiwifruit puree before high-pressure processing (HPP), after HPP, and during storage for 28 days at 4 ℃.

Sample		*η* = *k* ∙ *γ*^(*n*−1)^				G′ = *K*′ • *ω ^n^*′		G″ = *K*″ *• ω ^n^*″		
		*k*	*N*	*R^2^*	*K′*	*n*′	R^2^	*K*″	*n*″	R^2^
Orange juice	Control	83.97 ± 4.66	0.37 ± 0.03	0.92	0.00 ± 0.00	2.08 ± 0.34	0.65	0.01 ± 0.00	2.08 ± 0.04	0.99
	0	83.97 ± 4.66	0.37 ± 0.03	0.92	0.01 ± 0.00	2.08 ± 0.04	0.99	0.00 ± 0.00	2.07 ± 0.04	0.99
	28	42.3 ± 2.98	0.52 ± 0.04	0.74	0.00 ± 0.00	5.05 ± 1.53	0.42	0.00 ± 0.00	5.1 ± 0.08	1.00
Mixed juice	Control	1960.29 ± 16.44	0.08 ± 0	1.00	2.24 ± 0.29	0.00 ± 0.00	−0.02	0.02 ± 0.01	1.61 ± 0.09	0.95
	0	1685.11 ± 10.32	0.00 ± 0.00	1.00	2.81 ± 0.37	0.00 ± 0.00	−0.02	0.01 ± 0.01	1.91 ± 0.15	0.91
	28	3057.43 ± 39.43	0.17 ± 0.01	1.00	3.01 ± 0.49	0.00 ± 0.00	−0.04	0.00 ± 0.00	4.15 ± 0.08	1.00
Kiwi-fruit puree	Control	24520.41 ± 455.27	0.09 ± 0.01	1.00	56.83 ± 5.04	0.00 ± 0.00	−0.02	0.06 ± 0.02	1.78 ± 0.06	0.98
	0	24778.52 ± 364.16	0.12 ± 0.01	1.00	59.83 ± 6.04	0.00 ± 0.00	0.38	0.03 ± 0.02	1.64 ± 0.12	0.91
	28	28574.57 ± 285.76	0.18 ± 0.01	1.00	60.99 ± 5.62	0.00 ± 0.00	−0.04	0.00 ± 0.00	3.04 ± 0.34	0.89

## Data Availability

Data is contained within the article.

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
