# Peer review of "A Novel Strategy to Improve Cloud Stability of Orange-Based Juice: Combination of Natural Pectin Methylesterase Inhibitor and High-Pressure Processing"

_foods, 2023, doi:10.3390/foods12030581_

Round 1
Reviewer 1 Report
The article is interesting but some I have some remarks.
Chapter 2.4 Authors should add the gap value and the type of geometry used in frequency sweep test as well as shear tests.
Line 252: Chapter name should in next line.
It is not clear how Authors calculated the average viscosity, the viscosity depends on the shear rate for non-Newtonian fluids. Thus, it is possible to calculate consistency K or the determine viscosity at selected shear rate but not the average viscosity.
Chapter Conclusions: Some part of text should be moved to Chapter results and discussion. Now chapter Conclusion looks like the abstract, it is more statements than conclusions. This part needs improvement.
Reviewer 2 Report
The manuscript proposes a method to increase the cloud stability of orange juice treated by high hydrostatic pressure by adding kiwi puree due to the presence of pectin methylsterase inhibitors. In general, the article is well structured and easy to understand, but it requires improvements in some aspects.
The abstract is difficult to understand independently from the rest of the manuscript, especially regarding the purpose of using the kiwi puree. It should be explained that this is used because of the presence of PME inhibitors.
Pg 3 ln 102: the temperature at which the HHP treatment is carried out should be included, as well as the times in which the maximum pressure is reached and also that of depressurization. Also explain the type of container used to treat the juice
Pg 5 Ln 209: Specify how many replicates have been made of each type of sample and treatment for the statistical study of the data
Pg 7 Ln 267: It is mentioned that the juice has been homogenized, but this treatment is not mentioned in the Mat&Met section: describe the way in which the treatment has been carried out, including the temperature and pressure used
Consider reviewing figures 2, 3 and 5, as they include many graphs of different types, especially figure 3. Consider separating these graphs into different figures for better understanding
